# Towards Hard-pose Virtual Try-on via 3D-aware Global Correspondence Learning

**Zaiyu Huang**[1*], **Hanhui Li**[1*], **Zhenyu Xie**[1,2]
**Michael Kampffmeyer**[3], **Qingling Cai**[1†], **Xiaodan Liang**[1,4†]
[1]Shenzhen Campus of Sun Yat-Sen University
[2]ByteDance, [3] UiT The Arctic University of Norway
[4]Peng Cheng Laboratory
{huangzy225, xiezhy6} @mail2.sysu.edu.cn
{lihh77, caiqingl} @mail.sysu.edu.cn
michael.c.kampffmeyer@uit.no, xdliang328@gmail.com

## Abstract

In this paper, we target image-based person-to-person virtual try-on in the presence of diverse poses and large viewpoint variations. Existing methods are restricted in this setting as they estimate garment warping flows mainly based on 2D poses and appearance, which omits the geometric prior of the 3D human body shape. Moreover, current garment warping methods are confined to localized regions, which makes them ineffective in capturing long-range dependencies and results in inferior flows with artifacts. To tackle these issues, we present 3D-aware global correspondences, which are reliable flows that jointly encode global semantic correlations, local deformations, and geometric priors of 3D human bodies. Particularly, given an image pair depicting the source and target person, (a) we first obtain their pose-aware and high-level representations via two encoders, and introduce a coarse-to-fine decoder with multiple refinement modules to predict the pixel-wise global correspondence. (b) 3D parametric human models inferred from images are incorporated as priors to regularize the correspondence refinement process so that our flows can be 3D-aware and better handle variations of pose and viewpoint. (c) Finally, an adversarial generator takes the garment warped by the 3D-aware flow, and the image of the target person as inputs, to synthesize the photo-realistic try-on result. Extensive experiments on public benchmarks and our HardPose test set demonstrate the superiority of our method against the SOTA try-on approaches.

## 1 Introduction

Current image-based virtual try-on (VTON) methods typically include two key modules, namely a warping module that transfers a given garment to the desired pose and a blending module/generator that synthesizes the try-on result given the target person and the warped garment. To facilitate high-quality VTON, it is therefore crucial to accurately preserve garment details (e.g., color, texture, shape, pattern) in the warping stage. This, however, is a difficult problem, which most existing methods try to address by relying on thin plate splines (TPS) [1] or appearance flows [2].

While TPS is a well-studied and efficient method that performs the warping based on a set of control points and has a closed-form solution, it is prone to inaccurate transformations under large geometric deformations [3]. To further improve the warping quality, there has been a shift towards directly

---

*Both authors contributed equally.

†Corresponding authors. Our code will be available at 3D-GCL.

36th Conference on Neural Information Processing Systems (NeurIPS 2022).

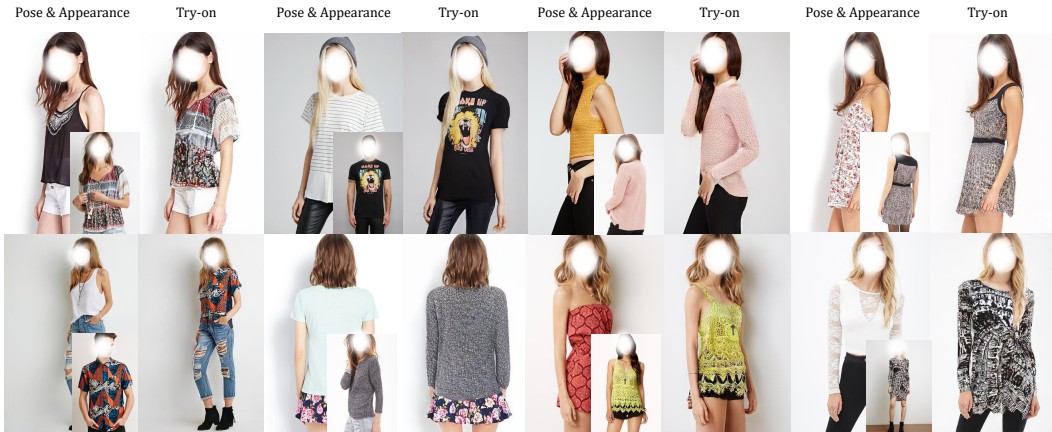

Figure 1: Visual examples of the proposed 3D-aware global correspondence learning method for person-to-person virtual try-on. Our method can accurately preserve garment textures and generate photo-realistic results under diverse poses and viewpoints.

estimating appearance flow fields by leveraging the capacity of deep neural networks. These methods perform well in relatively easy try-on scenarios, such as transferring flat, in-shop garments with pure colors or simple patterns to desired poses, but still suffer from severe artifacts in unconstrained scenes like person-to-person VTON, where varying poses and viewpoints commonly exist.

These artifacts are caused by the following two reasons: First, due to the lack of 3D geometric guidance in current methods, flows estimated merely based on appearance and 2D poses are unable to represent the deformable, non-rigid characteristics of garments, which results in texture distortions and improper warping. Second, for the purpose of efficiency, mainstream flow-based networks are usually constructed by stacking modules with limited receptive fields, e.g., ClothFlow [4] simply uses a convolutional layer to predict its initial cloth flow. As the effective area of a network module may be a small fraction of its theoretical receptive field [5], flows estimated by existing methods are de facto restricted to local areas and cannot leverage long-range dependencies effectively to tackle pose variations and occlusions.

Visual correspondences [6, 7] that depict cross-instance/domain semantic mappings could be a potential alternative to appearance flows for tackling pose and viewpoint variations, because they exploit correlations among high-level concepts, instead of appearance-oriented pixel-wise offsets. However, without extra guidance/regularization we observe that these approaches tend to focus on large and apparent structures (e.g., human pose) while ignoring subtle details and thus have a detrimental effect on the VTON user experience. Moreover, the complexity of calculating high-resolution correspondences directly is expensive ($O(N^2)$, where $N$ is the resolution), which further hinders its subsequent processes and downstream applications.

A feasible solution is to leverage 3D human models (e.g. SMPL [8]) to generate 3D-aware correspondences that are less ambiguous. However, a naive implementation that simply replaces the 2D pose representation in a network with its 3D counterpart is ineffective. This is because in our hard-pose VTON task, these 3D human models fail to predict the body mesh correctly and inevitably introduce noise to the estimated flow. Besides, employing 3D human models will lead to considerably lower inference speed [9].

Therefore, we propose a 3D-aware global correspondence learning (3D-GCL) method to tackle the aforementioned issues. The key idea of our 3D-GCL method is to leverage geometric priors of 3D human bodies to guide the correspondences, so that modelling non-rigid deformations of garments can be more effective. Unlike previous methods that incorporate 3D information into their inputs explicitly, we consider *the implicit supervision of 3D priors in our correspondence learning process*. In this way, not only our correspondences can be 3D-aware, but also the side effects of improper 3D information inferred from images can be alleviated. Specifically, given a source image and a target image, we first obtain their high-level and pose-aware feature maps via two parallel encoder networks, as well as the global correspondence via computing the correlation between the feature maps. After that, multiple geometry-aware correspondence refinement modules (GACRMs) are introduced to

upsample and refine the global correspondence stage-by-stage, thereby efficiently constructing the final full-resolution correspondences. More importantly, for each of the GACRMs, an elaborately designed 3D regularization loss is adopted to provide the GACRMs with the ability to leverage 3D geometric priors. At last, we warp the garment with the final correspondence, combine it with the image of the target person, and synthesize the try-on result via an adversarial generator.[3] Experiments on public benchmarks and our manually selected HardPose test set demonstrate that our 3D-GCL network synthesizes photo-realistic virtual try-on results that accurately preserve garment structures and textures even for hard-pose cases.

Our contribution can be summarized as follows:

- We elaborately design a novel 3D-aware global correspondence learning framework for virtual try-on in the presence of diverse pose variations and viewpoint changes.
- We present a geometry-aware correspondence refinement module and a 3D correspondence regularization loss, which together facilitate the understanding of geometric information and ensure the preservation of garment textures in our network.
- We conduct extensive experiments to validate the superiority of our method against state-of-the-art approaches, in terms of both garment warping and try-on synthesis.

## 2 Related Work

**Virtual Try-on** The most challenging aspect of VTON is the preservation of diverse textures and logos, which is addressed in current approaches by heavily relying on warping modules. One popular technique to warp the in-shop garment that has been employed by several early approaches [12, 13] is to utilize the TPS [1]. However, while these models can maintain garment details, they tend to introduce undesired distortions and artifacts under large pose deformations. To alleviate these artifacts, extra constraints and visual cues have been exploited [14, 15] to refine the control points of the TPS. However, these approaches are still limited by the sparse control-point-based representation of the TPS. Hence, there has been a shift to utilize networks to predict dense flow fields directly [4, 15, 16], which has considerably improved in-shop VTON performance. Nevertheless, as we have emphasized, existing methods are unable to precisely represent long-range and local deformations via a single flow, and hence they are restricted in the hard-pose VTON task.

**3D-aware Synthesis Models** Recent advances in 3D pose estimation and reconstruction (e.g., [8, 17, 18]) facilitate the use of 3D human models in synthesis methods [3, 9, 19, 20, 21]. Despite the effectiveness of 3D human models in modeling local dependencies, current 3D-aware synthesis models struggle to predict long-range deformations due to their unstable gradient propagation. This might be caused by inaccurate 3D model fitting, intertwined flow and feature gradients, and the use of bilinear sampling [22]. Contrary to existing methods, we first predict a global correspondence, and then augment it via the regularization provided by 3D human models. Hence we can circumvent the difficulty of learning long-range flows merely based on 3D human models.

**Correspondence Learning** Several cross-domain image synthesis methods based on correspondence learning [7, 20, 23, 24] have been proposed recently, among which [20] adopts the same pose representation as in the proposed method. However, its learned flows are restricted to a predefined UV space, which limits its scalability and performance. The proposed method instead does not include such restrictions, as its correspondences refined by 3D human models are robust to various poses and body shapes.

## 3 3D-aware Global Correspondence Learning

Our goal is to overcome the limitations of current try-on methods in handling large pose and viewpoint variations. This requires us to estimate flows that 1) can model long-range correspondences to tackle occlusions, rare poses, and novel views; and 2) can preserve textures and local structures of garments to ensure accurate try-on results. To this end, given a pair of person images, our proposed 3D-GCL

---

[3]Note, while [10] incorporates the 3D prior into the training of the network, the intention and the derivation of their 3D prior are different from our 3D-GCL. Similarly, while [11] utilizes the global information for flow estimation, it does not explicitly model the correspondence between the source and target feature. A detailed discussion of these two methods is included in the supplementary.

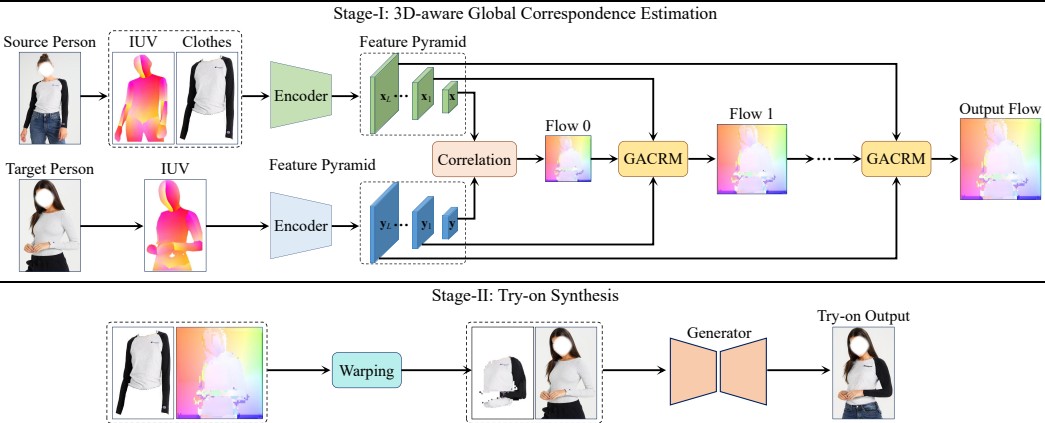

Figure 2: Overview of the proposed two-stage 3D-GCL framework. In the first stage, the 3D-aware global correspondence is predicted by the correlation between the pose-aware features extracted from the input images and is refined by multiple GACRMs. In the second stage, the target garment is warped by the 3D-aware global correspondence, and fed into a generator along with the image of the target person to generate the try-on result.

network (Sec. 3.1) first estimates a global flow via the correlation between their high-level 3D pose-aware features (Sec. 3.2), and then adopts multiple GACRMs to guide the global flow in depicting local deformations (Sec. 3.3). Our training objectives are provided in Sec. 3.4.

## 3.1 Network Architecture

Given an image $I_s$ of the source person wearing a garment $G$ and a target person image $I_t$, our task is to generate a photo-realistic try-on image where the target person is wearing $G$. As shown in Fig. 2, our 3D-GCL network tackles this task in a two-stage manner: In the first stage, we employ an efficient encoder-decoder architecture $f(I_s, I_t)$ to estimate the 3D-aware global correspondence for warping $G$, i.e., $f : \mathbb{R}^{H \times W \times 3} \times \mathbb{R}^{H \times W \times 3} \to \mathbb{R}^{H \times W \times 2}$, where $H$ and $W$ denote the image height and width. In the second stage, we adopt a conditional generator $g(I_t | G')$ to synthesize the final try-on result $I'_t$, where $G' \in \mathbb{R}^{H \times W \times 3}$ is the warped garment. Choices for realizing $g$ are flexible, and we simply employ a modified StyleGAN2 [25], where we replace the original style normalization strategy with spatially-adaptive normalization [26] to better maintain the spatial structures in garments. The details of our network architecture are provided in the supplementary material.

## 3.2 Pose-aware Global Correspondence Estimation

Without the proper guidance of 3D pose information, existing appearance flows often fail to distinguish body parts (e.g., torso and arms) under heavy occlusions and pose variations. Moreover, full-resolution visual correspondences are computationally expensive. Therefore, we propose to exploit features that are pose-aware and compact for correlation learning, so that these two issues can be tackled simultaneously.

Specifically, we first employ DensePose [17] on $I_s$ and $I_t$ to obtain their corresponding IUV maps $P_s, P_t \in \mathbb{R}^{H \times W \times 3}$, which are the pixel-wise estimations of a 3D parametric human body (i.e., SMPL [8]) projected on images. In contrast to previous appearance flow methods [2, 4], we then extract features directly from $P_s$ and $P_t$ rather than from the input images or segmentation masks.[4] This is reasonable, as the I channel and the UV channels of $P_s$ and $P_t$ have encoded semantic parts and positions of the 3D human body, respectively. Furthermore, this allows our network to disentangle irrelevant appearance information (e.g., faces and skin tones) and focus on pose and shape variations. Following the same motivation, we also leverage a human parser [27] to obtain the segmentation of the target garment $G$. Two parallel fully convolutional encoders are adopted for feature extraction, i.e., a source branch that takes the concatenation of $P_s$ and $G$ as input and a target branch with input $P_t$. Let $\mathbf{x}, \mathbf{y} \in \mathbb{R}^{N \times C}$ denote the output features of the source branch and the target branch,

---

[4]The ablation study for this choice is included in the supplementary material.

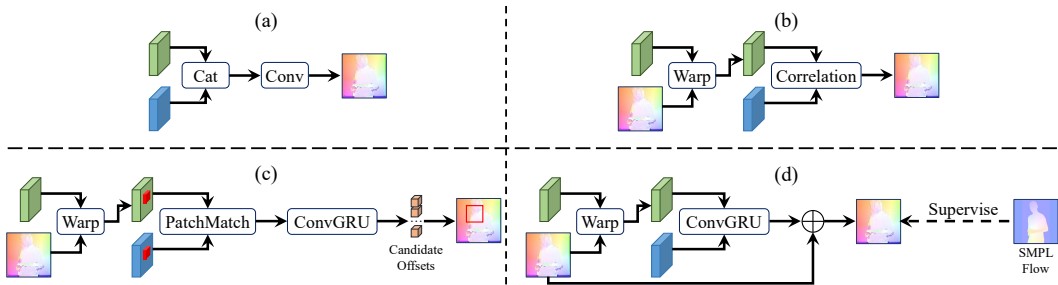

Figure 3: Comparison of flow estimation and refinement strategies. (a) Feature concatenation and convolution [4]. (b) Feature correlation [28, 15]. (c) GRU-assisted PatchMatch [24]. (d) The proposed GACRM. Flows computed from SMPL are exploited as the geometric prior for supervising the GACRM during the training phase, and consequently we can obtain precise flows without computationally expensive operations like high-resolution feature correlations and PatchMatch.

where $N = \frac{H}{S} \times \frac{W}{S}$ is the scaled spatial resolution and $C$ is the number of feature channels. $S = 8$ throughout this paper, so that $N \ll H \times W$, which ensures that the estimation of the correspondence is computationally efficient. We calculate the correlation between $\mathbf{x}$ and $\mathbf{y}$ as follows:

$$\mathbf{o} = \begin{bmatrix} \mathbf{x}_1\mathbf{y}_1 & \dots & \mathbf{x}_1\mathbf{y}_N \\ \vdots & \ddots & \vdots \\ \mathbf{x}_N\mathbf{y}_1 & \dots & \mathbf{x}_N\mathbf{y}_N \end{bmatrix}, \tag{1}$$

where $\mathbf{o}_{ij}, 1 \leq i, j \leq N$ measures the semantic and pose similarity between the $i$-th feature vector in $\mathbf{x}$ and the $j$-th feature vector in $\mathbf{y}$. As $\mathbf{o}$ contains the similarities between all possible spatial combinations of feature vectors in $\mathbf{x}$ and $\mathbf{y}$, it is able to capture long-range variations. Hence we consider $\mathbf{o}$ as our global correspondence. A coarse warping result can be obtained via the linear combination of $G$ weighted by $\mathbf{o}$ directly, i.e., we can down-sample $G$ and reshape it to $N \times 3$, and then obtain $G'$ as $G' = \mathbf{o}G$. However, the down-sampling process will damage the quality of $G$ severely. Therefore, we present the GACRM in the next section, which generates the full-resolution and accurate flow based on the global correspondence.

**Comparison with other correlation based methods**. Several recently proposed methods [7, 24] also consider feature correlation for correspondence estimation. However, they target image translation across different domains (e.g., from sketches to photos) while our focus is on estimating global correspondences to warp garments (from poses to poses). Besides, they rely on a 2D keypoint-based representation of poses [29] that is sparse and cannot represent body shapes, while we adopt the 3D dense representation of poses that leads to more robust estimated correspondences. More importantly, our estimated global correspondences serve as the base flows and are refined by GACRMs to tackle both long-range and local pose variations.

### 3.3 Geometry-aware Correspondence Refinement

A potential issue of correlation based global appearances is that they ignore the spatial structures of features (as in Eq. (1)). Consequently, it is hard to ensure local consistency in the global appearances. Even worse, the correlation between high-level semantic features tends to generate over-smooth flows that cannot preserve garment details such as patterns and logos. Therefore, we present the GACRM and a 3D regularization loss to leverage the geometric prior of 3D human bodies and tackle these issues.

To begin with, we convert $\mathbf{o}$ into a flow $\mathbf{f}_0$ of size $\frac{H}{S} \times \frac{W}{S}$ via applying the argmax function. To incorporate local details into $\mathbf{f}_0$, we exploit the early-stage feature pyramids in both the source branch and the target branch. Assume both feature pyramids consist of $L$ layers (as shown in Fig. 2), we utilize a GACRM for each layer, which can be formulated as follows:

$$\mathbf{f}_{l+1} = \mathbf{f}_l^u + h(\mathbf{x}_{l+1}, \mathbf{y}_{l+1}, \mathbf{f}_l^u), \tag{2}$$

where $0 \leq l < L$ is the layer index. $\mathbf{x}_l$ and $\mathbf{y}_l$ denote feature maps of the $l$-th layer of the source and target feature pyramids and $\mathbf{f}_l^u$ is the up-scaled flow from the previous layer. $h$ is responsible for

predicting a residual flow to enhance $\mathbf{f}_l^u$, which is realized simply by a convolutional gated recurrent unit [24] (ConvGRU) taking $\mathbf{x}_l$ warped by $\mathbf{f}_l^u$ and $\mathbf{y}_l$ as inputs, as shown in Fig. 3.

To incorporate the geometric prior of 3D human bodies into our correspondence refinement process, we propose a novel 3D regularization loss for the global correspondence and its corresponding refined flows. Specifically, we adopt an off-the-shelf 3D mesh regressor [30] to obtain the SMPL models of the source person and the target person. Following [9], we render the two SMPL models to obtain a visibility mask, and calculate a SMPL flow measuring the disparity between the source person mesh and the target person mesh projected on the 2D image plane. The SMPL flow is utilized as the pseudo ground-truth in our 3D regularization loss, which is defined as follows:

$$\mathcal{L}_r = \mathcal{L}_o + \mathcal{L}_f, \tag{3}$$

where $\mathcal{L}_o$ and $\mathcal{L}_f$ are loss terms for the global correspondence and the refined flows. In the remainder of this section, for the conciseness of notation and presentation, we assume that all related variables in the 3D regularization loss are resized accordingly. To define $\mathcal{L}_o$, note that it is common practice to conduct interpolations (e.g., bilinear sampling) in flow based warping, which are similar to the aforementioned coarse warping based on the global correspondence. Therefore, we generate a pseudo ground-truth correspondence by scattering the interpolation weights of each pixel to its corresponding position in the correlation matrix, making the image warped by our pseudo ground-truth correspondence visually close to its counterpart warped by the SMPL flow. Formally, we obtain the ground-truth correspondence $\mathbf{o}^{gt} \in \mathbb{R}^{N \times N}$ as follows:

$$\mathbf{o}_{ij}^{gt} = \begin{cases} w_j, & if \quad j \in \Omega(i) \quad and \quad G_i^{gt} = \sum_{k \in \Omega(i)} w_k G_k \\ 0, & otherwise \end{cases} \tag{4}$$

where $\Omega(i)$ denotes the reference points given $i$ as the target interpolation point, $w_j$ is the interpolation weight of the reference point $j$, and $G^{gt}$ denotes the ground-truth warped garment. We then define $\mathcal{L}_o$ as follows:

$$\mathcal{L}_o = \left\| M \odot (\mathbf{o}^{gt} - \mathbf{o}) G \right\|_1, \tag{5}$$

where $M \in \{0,1\}^{N \times N}$ is a binary mask with $M_{ij} = [\mathbf{o}_{ij}^{gt} > 0]$. $[\cdot]$ and $\odot$ denote the Iverson bracket and element-wise product, respectively. For $\mathcal{L}_f$, we consider the $l_1$ loss between the refined flows and the SMPL flow as follows:

$$\mathcal{L}_f = \sum_{l=1}^{L} \left\| V \odot (\mathbf{f}^{gt} - \mathbf{f}_l) \right\|_1, \tag{6}$$

where $V \in \{0,1\}^{H \times W}$ and $\mathbf{f}^{gt}$ denote the visibility mask and the SMPL flow, respectively.

The above 3D regularization loss encourages our network to predict flows that agree well with geometric deformations of 3D human bodies. This avoids the need of introducing complicated modules for imposing local constraints on flows, such as PatchMatch [24], local attention [22], or multi-stage correlations [15]. Consequently, we can adopt an elegant and efficient architecture for the GACRM, while obtaining pixel-wise and precise global correspondences.

### 3.4 Objective Functions

Since the 3D-GCL network consists of two stages, we train its correspondence estimation subnetwork in the first stage and the generator for try-on synthesis in the second stage separately. In addition to the proposed 3D regularization loss $\mathcal{L}_r$, we follow [20, 25] and use the $l_1$ loss $\mathcal{L}_{l_1}$, the perceptual loss $L_{perc}$, and the adversarial loss $\mathcal{L}_{adv}$ for training. The total loss for the correspondence estimation stage $\mathcal{L}_c$ is therefore defined as follows:

$$\mathcal{L}_c = \lambda_r \mathcal{L}_r + \lambda_{l_1}^c \mathcal{L}_{l_1}^c + \lambda_{perc}^c \mathcal{L}_{perc}^c, \tag{7}$$

where $\lambda_r$, $\lambda_{l_1}^c$, and $\lambda_{perc}^c$ are weighting hyperparameters. $\mathcal{L}_{l_1}^c$ and $\mathcal{L}_{perc}^c$ are applied to the garments warped by $\mathbf{f}_0, ..., \mathbf{f}_L$. The total loss for the generator $\mathcal{L}_g$ is defined as follows:

$$\mathcal{L}_g = \lambda_{adv} \mathcal{L}_{adv} + \lambda_{l_1}^g \mathcal{L}_{l_1}^g + \lambda_{perc}^g \mathcal{L}_{perc}^g, \tag{8}$$

where $\lambda_{adv}$, $\lambda_{l_1}^g$, and $\lambda_{perc}^g$ are hyperparameters.

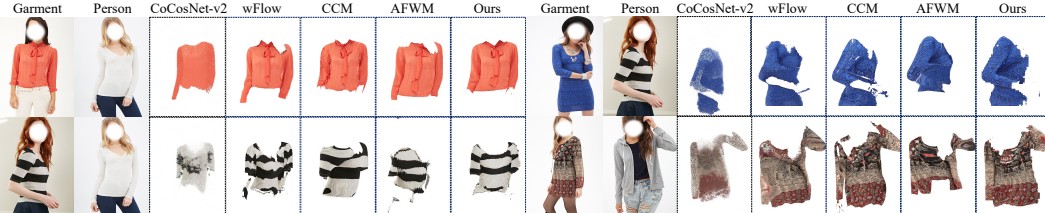

Figure 4: Visual comparison of clothes warped by our 3D-aware global correspondences and other state-of-the-art warping methods. Our method better preserves garment details and structures under diverse poses and viewpoints.

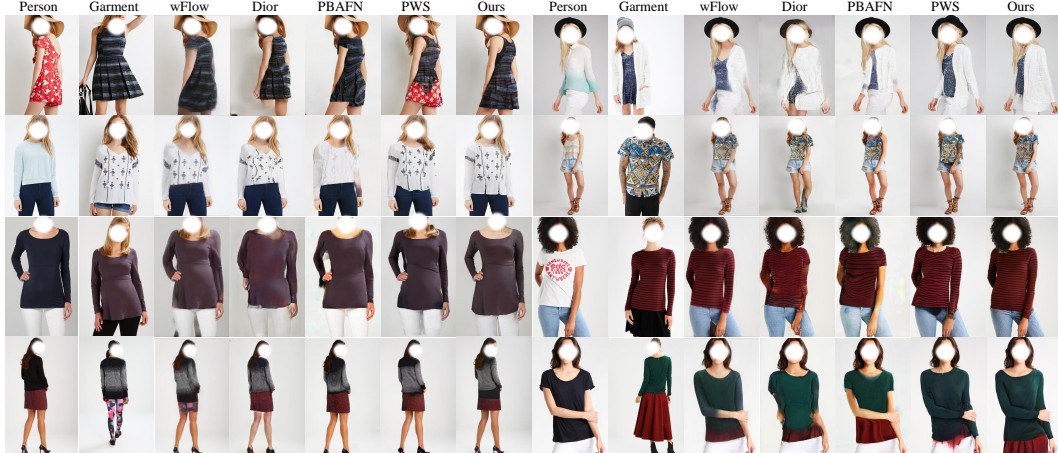

Figure 5: Visual comparison of our method and state-of-the-art VTON approaches. Our method is more robust to hard-pose cases and synthesizes photo-realistic try-on results.

## 4   Experiments

In this section, we conduct extensive experiments to validate the effectiveness of the proposed 3D-GCL network. Due to the page limitation, the interested reader is refer to the supplemental material for more experimental settings and results.

**Datasets** Our experiments are conducted on two open-source datasets, DeepFashion [31] and MPV [32], which contain 52,712 and 37,723 fashion images, respectively. To ensure a fair comparison, we follow the train/test split of [20, 22, 33] on the DeepFashion dataset and use the original split of MPV. Additionally, pairs without properly detected IUV maps or SMPL models are filtered out. In the test phase, query persons on the test list are further shuffled to match the task setting of VTON. In this way, we get 101,622/8,564 train/test pairs for DeepFashion and 52,236/10,544 train/test pairs for MPV. Furthermore, as this paper targets hard-pose VTON, we manually select test image pairs from DeepFashion and MPV to construct the HardPose test set. The HardPose test set consists of 2,406 image pairs, where each test pair contains diverse pose changes or large viewpoint variations. The details and visual examples of this HardPose test set can be found in the supplemental material.

**Evaluation Metrics** We evaluate the performance of both garment warping and final try-on synthesis. For the warping stage, we apply LPIPS [34] to evaluate the similarity between the warped garments and the corresponding ground-truths extracted by the human parser [27]. We also adopt the mean Intersection over Union (mIoU) metric to evaluate the semantic-correctness of the estimated flows by comparing the segmentation maps of the warped garments with their ground-truths. Furthermore, following [14, 15, 21, 35], we perform a human evaluation by inviting 40 volunteers to select the warped clothes with the best visual quality among our method and the baseline approaches.

For the try-on stage, note that ground-truth try-on results are not available. Thus we select the widely used unpaired Fréchet Inception Distance (FID) to quantify the quality of the synthesis results. We also perform a human evaluation study following recent virtual try-on methods [14, 15, 21]. 40 volunteers are invited to fill in a questionnaire, in which they are presented with the try-on results of all methods and are asked to select the most realistic one.

| Method | FullSet | | | | HardPose | | | | | |
|---|---|---|---|---|---|---|---|---|---|---|
| | DeepFashion | | MPV | | DeepFashion | | | MPV | | |
| | mIoU ↑ | LPIPS ↓ | mIoU ↑ | LPIPS ↓ | mIoU ↑ | HE ↑ | LPIPS ↓ | mIoU ↑ | HE ↑ | LPIPS ↓ |
| AFWM [15] | 71.24 | **0.1656** | 74.56 | **0.1012** | 63.36 | 25.63% | **0.1648** | 66.63 | 32.50% | **0.1014** |
| CoCosNet-v2 [24] | - | 0.2269 | - | 0.1873 | - | 2.375% | 0.2251 | - | 3.750% | 0.1800 |
| CCM [20] | 62.74 | 0.1997 | 66.75 | 0.1655 | 65.17 | 16.75% | 0.1865 | 63.92 | 8.625% | 0.1575 |
| wFlow [21] | - | 0.1686 | - | 0.1314 | - | 13.38% | 0.1699 | - | 10.00% | 0.1302 |
| 3D-GCL(Ours) | **75.76** | 0.1725 | **75.46** | 0.1318 | **74.22** | **41.88%** | 0.1732 | **71.89** | **45.13%** | 0.1345 |

Table 1: Comparison of warped clothes produced by different methods.

| Method | FullSet | | | | HardPose | | | |
|---|---|---|---|---|---|---|---|---|
| | DeepFashion | | MPV | | DeepFashion | | MPV | |
| | FID ↓ | HE ↑ | FID ↓ | HE ↑ | FID ↓ | HE ↑ | FID ↓ | HE ↑ |
| PBAFN [15] | 21.49 | 7.625% | 10.62 | 12.50% | 22.34 | 7.833% | 14.35 | 14.67% |
| Dior [36] | 26.58 | 5.375% | 26.62 | 3.875% | 27.49 | 3.500% | 30.70 | 4.500% |
| Pose with style [20] | 16.35 | 27.38% | 6.78 | 20.88% | 18.50 | 26.17% | 9.808 | 19.83% |
| wFlow [21] | 19.81 | 11.63% | 12.81 | 13.25% | 22.97 | 9.667% | 20.41 | 11.33% |
| 3D-GCL (Ours) | **10.58** | **48.00%** | **6.00** | **49.50%** | **11.37** | **52.83%** | **8.776** | **49.67%** |

Table 2: Comparison of final try-on results produced by different methods.

**Implementation Details** The proposed 3D-GCL network is implemented in PyTorch and trained with 4 Tesla V100 GPUs. We first train the correspondence estimation subnetwork for 20 epochs with a batch-size of 8, and then follow the settings of [20, 25] to train the try-on generator.

**Baselines** We compare the warping and try-on performance of our model with publicly-available state-of-the-art methods. For the warping stage, we leverage the FlowNet backbone [28] for all methods and consider the warping models of AFWM [15], CCM [20], and wFlow [21] as the baseline methods. For the evaluation on VTON, we select four cutting-edge methods, including PBAFN [15], Dior [36], Pose-with-Style [20], and wFlow [21].

## 4.1 Qualitative results

As demonstrated in Fig. 4, conventional warping strategies adopted in PBAFN [15] and Dior [36] fail to capture correct correspondences in images. Although they employ an extra smoothness constraint [15], their warped garments are often violently distorted to fill the target silhouette facing complex poses and view-point changes. Moreover, they also ignore 3D information such as orientations and body geometry, and thus harm the visual quality of their warped garments. The hybrid approaches Pose-with-Style [20] and wFlow [21] struggle to fuse different flow-based warping methods in a harmonious way. wFlow produces considerably blurry results since its composition masks are unreliable in low confidence regions. Pose-with-Style on the other hand only warps pixels in the pre-defined UV space of DensePose, which limits its application for diverse garment types such as skirts and dresses. On contrary, our method produces favourable results even in challenging scenarios with large pose and viewpoint changes. Note that with the help of global correspondence learning, our method has a larger receptive field and is able to also generate more compelling results in front view transformations. Fig. 5 illustrates the visual comparison of our try-on results with the baseline methods. As mentioned in Section 1, the try-on results are significantly affected by the quality of the deformed garments. Our method, with more accurate and harmonious estimated correspondences, outperforms the other four baseline methods in visual quality considerably.

## 4.2 Quantitative results

The quantitative comparison between our correspondence estimation method against other baselines is reported in Table 1. Our method outperforms existing advanced warping methods such as CoCosNet-v2 [24] and CCM [20], and achieves comparable performance to wFlow [21] on the LPIPS metric. Note that AFWM [15] obtains considerably lower LPIPS scores. This is mainly because AFWM adopts the expensive multi-stage feature correlations and encourages extremely smooth flows. Yet such a strategy tends to ignore the spatial and semantic structures in garments, and consequently yields inferior visual quality in complex scenarios. To validate this, we also measure the mIoU metric and assess the human evaluation (HE) score. As reported in Table 1, our method obtains the highest mIoU scores, indicating that our warped garments are more semantically accurate. Our method also

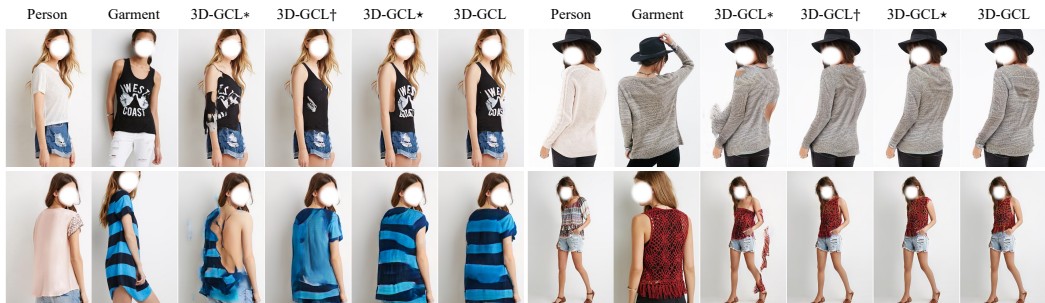

Figure 6: Visual comparison of the proposed 3D-GCL network with different configurations.

| Method | Global Correspondence | 3D Regularization Loss | mIoU ↑ | FID ↓ |
|---|---|---|---|---|
| 3D-GCL ∗ | ✗ | ✗ | 50.21 | 17.25 |
| 3D-GCL † | ✓ | ✗ | 57.36 | 11.53 |
| 3D-GCL ⋆ | ✗ | ✓ | 62.06 | 10.98 |
| 3D-GCL | ✓ | ✓ | **74.22** | **10.58** |

Table 3: Ablation study of the proposed 3D-GCL network on DeepFashion [31].

obtains the highest human evaluation percentage, which indicates that it produces the best visual quality for warped garments under large viewpoint changes and pose variations.

As shown in Table 2, our method also outperforms other state-of-the-art VTON methods persistently across different test sets. Our method obtains the lowest FID scores, which indicates that our method can produce high-fidelity try-on results. This is also validated through the human evaluation. Note that most of the presented methods perform better on the MPV dataset than on the DeepFashion dataset. This is because images on the MPV dataset contain less pose variations and viewpoint changes. On the DeepFashion dataset, our method still surpasses the second best method (Pose-with-Style) by a considerable margin, which suggests that our method can generalize well in complex scenarios.

### 4.3   Ablation Study

To validate the effectiveness of the global correspondence estimation strategy and the 3D regularization loss, we implement three variants of the proposed method and evaluate them on DeepFashion. The experimental results of this ablation study is reported in Table 3.

**Effect of Global Correspondence.** Compared with the vanilla variant of the proposed 3D-GCL network (3D-GCL ∗), adopting global correspondences (3D-GCL †) provides us with the performance gains persistent on two metrics. This is expected, as global correspondences are effective in capturing long-range deformations.

**Effect of 3D Regularization Loss.** Compared with global correspondences, the advantages of the 3D regularization loss (3D-GCL ⋆) are even more significant, especially according to the mIoU metric. As discussed in Sec. 3.3, correspondence learning only matches pose features semantically, which means that, on their own, they are unable to preserve fine-grained structures and textures. This result suggests that the 3D regularization loss leveraging SMPL flows to provide geometric guidance, is a necessary and effective component of our network.

Finally, our full model obtains the best performance compared with other variants, which further supports that flows encoding both long-range and local deformations are critical in producing compelling try-on results in challenging scenarios. The visual comparison of different variants of our method is shown in Fig. 6.

## 5   Conclusion

In this paper, we propose a 3D-aware global correspondence learning (3D-GCL) network to tackle hard-pose person-to-person visual try-on. Our 3D-GCL network adopts an efficient architecture for global correspondence estimation and refinement, and introduces a novel 3D regularization loss to exploit geometric priors of 3D human bodies to guide the refinement process of the global

correspondences. In this way, our full-resolution 3D-aware global correspondences can represent both long-range and local deformations precisely, and hence our method can handle large pose and viewpoint variations, while preserving garment textures and structures effectively. Quantitative and qualitative comparison validate the effectiveness of the proposed 3D-GCL network. Due to its effectiveness and flexibility, the proposed 3D-GCL network constitutes a new baseline for hard-pose try-on, and more advanced components, like attention models, can be integrated into the network in the future for further improvements.

## 6 Acknowledgement

This work was supported in part by National Key R&D Program of China under Grant No. 2020AAA0109700, National Natural Science Foundation of China (NSFC) under Grant No.U19A2073, No.U1811461, No.61902088 and No.61976233, Shenzhen Fundamental Research Program (Project No. RCYX20200714114642083, No. JCYJ20190807154211365) and CAAI-Huawei MindSpore Open Fund. We thank MindSpore for the partial support of this work, which is a new deep learning computing framwork[5].

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
