# OpenReview forum: "Towards Hard-pose Virtual Try-on via 3D-aware Global Correspondence Learning"
_NeurIPS.cc/2022/Conference — NeurIPS 2022 Accept_

### Official Review · Reviewer_bUCs · 2022-07-11

**Rating:** 5
**Confidence:** 3
**Soundness:** 2 fair
**Presentation:** 3 good
**Contribution:** 2 fair

**Summary:**

This work proposes a virtual try-on technique to transfer garment from a given source image to target image/pose. A main difference to existing works is the use of dense correspondences between source and target poses and also using SMPL based regularization loss functions. Experiments on DeepFashion and MVP datasets with the proposed test splits demonstrate better results than existing techniques.

**Questions:**

- The results reported in Tables 1 and 2 do not match to those reported in earlier works. I have checked some of the compared papers and their reported numbers seem different. Does this work use different test split compared to existing works?

**Limitations:**

There is no clear discussion of limitations of the proposed technique. For instance, the proposed technique fails when the dense pose estimation on either source or target images fail.

**Strengths And Weaknesses:**

Strengths:
- SMPL based regularization losses for dense correspondence estimation.
- Better results on two different datasets compared to existing works on the proposed test splits.

Weaknesses:
- The main technical novelty of this work compared to prior art is the use of SMPL estimates for correspondence loss functions. As such the technical novelty is somewhat incremental.
- Line 224 says that "image pairs without properly detected IUV maps or SMPL models are filtered out". This seems unfair filtering of test set as the proposed method relies on IUV map and SMPL model estimation. As a result, the reported numbers could be biased towards the proposed technique. Also it is not clear what it means by 'properly' here.
- The details on how the 'HardPose' dataset is constructed is not clear. What constitutes a hard pose? Having some visual examples that distinguishes normal and HardPose test sets would be good. And, I do not find any extreme poses in any of the result images. So, it is not clear to me whether any of the images can be considered a hard pose.


Post rebuttal:
Authors addressed some of the concerns, especially w.r.t. experimental setup, in their response.

---

> ### Author Response · Authors · 2022-08-02
> **Response to Reviewer bUCs Part 2**
>
> **Detail of HardPose**
> Thanks for pointing out that the definition of "HardPose" in the paper might not be clear to the readers. We will elaborate our definition and also add some visual examples to the paper in the revision. In this paper, we define "HardPose" as opposite to standard posture, which is described as face forward and hands down. Pairs containing large viewpoint contrast or that have large differences in hand position or orientation between the source and target are regarded as "HardPose".
> We first employ the calculating mechanism for pose complexity in [5] to filter out easy samples, then manually pick out the pairs that contain visually "Hard" posture from the testing set to ensure a difficult subset.
> We compute the complexity score(lower is more complex) of this testing set, getting 74.39 for Deepfashion and 63.45 for MPV on the Hardset, compared to 77.87 for Deepfashion and 70.09 for MPV on the Fullset. We also include some visual examples of our HardPose testing set as well as an easy to hard example in the supplementary.
>
> **Mismatch of the reported results**
> Note that most of the baseline methods are implemented under different settings or leverage different train/test splits in the original papers. To ensure a fair comparison, all of the included methods in the tables are in our work compared using the train/test split of [1].
> The implementation of the evaluation metric may also cause slight differences as well. Here we use [6] to compute the distance between the distribution of our generated images and training data. Moreover, experiments designed in [1] are for pose transfer, while we conduct experiments following the common settings of Virtual Try-on, leading to different results even with the same train/test split. To further validate the effectiveness of our proposed model, we have now also compared our method with [1] under their Pose Transfer setting. Specifically, we swap top clothes of the same person in different poses in the images following the original test list of [1], getting 8,570 synthesis results. Then we compute the FID score of the generated images, obtaining a FID score of 9.53 for our method compare to [1]'s result of 13.57 (lower is better).
>
> ---
>
> [1] Pose with Style: Detail-Preserving Pose-Guided Image Synthesis with Conditional StyleGAN. ACM Transactions on Graphics 2021
>
> [2] ZFlow: Gated Appearance Flow-based Virtual Try-on with 3D Priors. ICCV 2021
>
> [3] Liquid Warping GAN: A Unified Framework for Human Motion Imitation, Appearance Transfer and Novel View Synthesis. ICCV 2019
>
> [4] Parser-Free Virtual Try-on via Distilling Appearance Flows. CVPR 2021
>
> [5] Towards Photo-Realistic Virtual Try-On by Adaptively Generating↔Preserving Image Content. CVPR 2020
>
> [6] On Buggy Resizing Libraries and Surprising Subtleties in FID Calculation. arXiv 2021

---

> ### Author Response · Authors · 2022-08-02
> **Response to Reviewer bUCs Part 1**
>
> Thank you for your careful revision and for acknowledging the good experimental results of our proposed methodology. In the following, we respond to your questions:
>
> * **Innovation of proposed method**: The main technical novelty of this work compared to prior art is the use of SMPL estimates for correspondence loss functions. As such the technical novelty is somewhat incremental.
> * **Unfair experiment setup**: Line 224 says that "image pairs without properly detected IUV maps or SMPL models are filtered out". This seems unfair filtering of test set as the proposed method relies on IUV map and SMPL model estimation. As a result, the reported numbers could be biased towards the proposed technique. Also it is not clear what it means by 'properly' here.
> * **Detail of HardPose**: The details on how the 'HardPose' dataset is constructed is not clear. What constitutes a hard pose? Having some visual examples that distinguishes normal and HardPose test sets would be good. And, I do not find any extreme poses in any of the result images. So, it is not clear to me whether any of the images can be considered a hard pose.
> * **Mismatch of the reported results**: The results reported in Tables 1 and 2 do not match to those reported in earlier works. I have checked some of the compared papers and their reported numbers seem different. Does this work use different test split compared to existing works?
>
> **Innovation of proposed method**
> We would like to emphasize that the technical contributions of this paper are two-fold: (i) The first one is indeed the SMPL supervised correspondence learning. However, in contrary to existing methods [1,2,3] that use SMPL flows as inputs, we only consider SMPL flows for supervising our correspondence learning. Such a strategy allows us to leverage geometric priors of 3D human bodies without fitting SMPL models in inference. (ii) More importantly, we have devised an efficient and effective network architecture for **global correspondence estimation** (Sec. 3.2 \& 3.3).
> Unlike other correspondence estimation methods, our proposed architecture can generate global correspondences (full resolution) without the expensive feature correlations/PatchMatch operations in existing methods (illustrated in Fig. 3 and Sec. 3.2, L157-L164).
> The superiority of the proposed architecture is further validated in our experiments as well.
>
> **Unfair experiment setup**
> Thanks for pointing out that this statement may cause confusion with regards to the experiment setup. We have elaborate this statement in the revised version and would like to ensure that the setup does not lead to a unfair experimental setup.
> Firstly, we only filter out the cases where no human is detected at all by DensePose and SMPL as these can be considered as outliers when regarding the global distribution of the data. We would like to stress that this process only filters out **345 of the original 101,967 pairs in the training set** and **6 of the original 8,570 pairs in the testing set** for Deepfashion, while no data is filtered out for the MVP dataset. Such a ratio of test pairs (6/8,570) has a negligible effect on the overall performance. To validate this quantitatively, we report the result of our method and the closest competitor [1] (Pose with Style) on the full testing set of Deepfashion. The experimental results are reported in Table 2 in the supplemental material and as follows:
>
> | Method | mIoU (the higher the better) | LPIPS (the lower the better) | FID (the lower the better) |
> | :-: | :-: | :-: | :-: |
> | Pose with Style (filtered) | 62.74 | 0.1997 | 16.35
> | Pose with Style (full) | 62.49 | 0.1998 | 16.35
> | 3D-GCL (Ours, filtered) | 75.76 | 0.1725 | 10.58 |
> | 3D-GCL (Ours, full) | 75.91 | 0.1725 | 10.58 |
>
> It is apparent from the above results that the effects of the 6 filtered out test pairs can be ignored, and the proposed method is superior to the baseline method in both cases.
>
> Secondly, the closest competing baseline method [1] and also [4] heavily rely on the DensePose results and will thus benefit from this process as well, leading to a fair comparison.

---

> ### Author Response · Authors · 2022-08-09
> **Response to Reviewer bUCs Part 3**
>
> Dear Reviewer bUCs,
> We have tried to address your concerns in our earlier responses (Part 1 and Part 2), and revised our paper based on your insightful suggestions. If you have any additional questions or suggestions, we would be very grateful to discuss them with you.

---

> > ### Comment · Reviewer_bUCs · 2022-08-09
> > **Thanks for the clarifications**
> >
> > Thanks authors for the clarifications. I will update my rating after cross-checking prior art on datasets and metrics.

---

> > > ### Comment · Reviewer_bUCs · 2022-08-10
> > > **Follow-up**
> > >
> > > Authors addressed some of my concerns, especially w.r.t. the issues in the experimental setup. I am increasing my rating to borderline accept.

---

### Official Review · Reviewer_nqeh · 2022-07-12

**Rating:** 6
**Confidence:** 4
**Soundness:** 3 good
**Presentation:** 3 good
**Contribution:** 3 good

**Summary:**

The paper proposes to reconstruct 3D human mesh from multiview images by fusing features with a transformer and assisting the training by pose alignment between different views.

**Questions:**

See weakness

**Limitations:**

yes

**Strengths And Weaknesses:**

Strengths
- The writing of the paper is clear.
- The motivation of the paper and the effectiveness of the proposed 3D-aware global correspondence is well supported by extensive experiments.

Weaknesses

More ablation studies could be added to analyze the effectiveness of the components of the method.
- How does the multiscale/coarse-to-fine structure influence the final result?
- Can the off-the-shelf 3D mesh regressor be applied to (a)(b)(c) in Figure 3?  Will the SMPL flow supervision benefit these strategies as well? How does the module structure affect the warping compared with other module structures?

---

> ### Author Response · Authors · 2022-08-02
> **Response to Reviewer nqeh**
>
> We thank the reviewer for the constructive feedback and for acknowledging the clear motivation, the efficient method, our extensive experiments and the clarity of the presentation. We are delighted to address the concerns and questions raised by the reviewer in the following:
>
> * **Influence of multiscale/coarse-to-fine structure**: How does the multiscale/coarse-to-fine structure influence the final result?
> * **More ablation study experiment results:** Can the off-the-shelf 3D mesh regressor be applied to (a)(b)(c) in Figure 3? Will the SMPL flow supervision benefit these strategies as well? How does the module structure affect the warping compared with other module structures?
>
> **Influence of multiscale/coarse-to-fine structure**
>
> | Method | mIoU (the higher the better) | LPIPS (the lower the better) | FID (the lower the better) |
> | :-: | :-: | :-: | :-: |
> | 3D-GCL w/o multiscale | 64.93 | 0.1968 | 12.70 |
> | 3D-GCL | **75.76** | **0.1725** | **10.58** |
>
> As discussed in the introduction and Sec.3.2, flow-based warping optimization tends to fall into local minima due to the limited receptive field, while the multi-scale structure helps facilitate the learning of the intermediate feature representations and thus promotes the overall performance of the model.
>
> **More ablation study experiment results**
>
> | Method | mIoU (the higher the better) | FID (the lower the better) |
> | :-: | :-: | :-: |
> | Structure-a: Feature convolutions | 68.57 | 11.65 |
> | Structure-b: Feature correlations | 72.04 | 11.74 |
> | 3D-GCL | **74.22** | **10.58** |
>
> Yes, the 3D mesh regressor can also be applied to (a) and (b), but not to (c) (because structure (c) does not estimate flows explicitly). We implemented two more variants of our method by replacing our GACRMs with the structure of (a) and (b). We evaluated them on the Deepfashion dataset, getting 11.65 FID score and 68.57 mIoU for structure (a), compared to 11.74 FID score and 72.04 mIoU for structure (b). For comparison, performance improves when leveraging GACRM to an FID of 10.58 and a mIoU of 74.22.

---

### Official Review · Reviewer_7wqS · 2022-07-16

**Rating:** 5
**Confidence:** 4
**Soundness:** 3 good
**Presentation:** 3 good
**Contribution:** 2 fair

**Summary:**

This work presents a 3D-aware global correspondence learning (3D-GCL) framework to tackle the image-based person-to-person virtual try-on problem. The core idea is to incorporate the geometric prior of 3D human body to guide the correspondence learning between the source and the target person, aiming at preserving detailed garment textures. To circumvent the difficulty of learning long-range correspondence, the authors introduce a coarse-to-fine framework. The garment warping flow is initialized via the global correlation between high-level image features and then progressively refined. In addition, the SMPL flow estimated from the source and the target person is introduced as the 3D-aware supervision signal to guide the refinement process. Empirical studies on the DeepFashion and MPV dataset demonstrate superior performance over previous methods, especially on hard pose samples.

**Questions:**


- Typos: Line 76 summarize --> summarized
- What are the selection criteria for the HardPose test set? Is there a quantitative analysis of how hard the HardPose test set is compared to the public benchmarks?
- Abalation study Table 3. What is the framework like for 3D-GCL w/o global correspondences?

**Limitations:**

Yes. The authors addressed the limitations and potential negative societal impact of their work.

**Strengths And Weaknesses:**

## Strength

- The proposed method is simple and effective. It demonstrates solid improvement over previous methods in terms of FID and human evaluation scores.

## Weakness

- Missing comparisons. Some recent works also focus on incorporating 3D priors [1] or global correspondence [2] into image-based virtual try-on. There is no deep discussion on the difference between this work and [1] [2]. Quantitative comparisons with [1][2] are also missing. Comparisons with the above works are needed to elucidate the main contributions of this work.

- Unfair experimental setup. Line 224. "pairs without properly detected IUV maps or SMPL models are filtered out." I am concerned whether this is a fair comparison with other methods, as filtering out failure detections is beneficial for testing the proposed method. This makes the empirical results less convincing.

[1] ZFlow: Gated Appearance Flow-based Virtual Try-on with 3D Priors. ICCV 2021

[2] Style-Based Global Appearance Flow for Virtual Try-On. CVPR 2022

---

> ### Author Response · Authors · 2022-08-02
> **Response to Reviewer 7wqS Part 2**
>
> **Unfair experimental setup**
> Thanks for pointing out that this statement may cause confusion with regards to the experiment setup. We have elaborate this statement in the revised version and would like to ensure that the setup does not lead to a unfair experimental setup.
> Firstly, we only filter out the cases where no human is detected at all by DensePose and SMPL as these can be considered as outliers when regarding the global distribution of the data. We would like to stress that this process only filters out **345 of the original 101,967 pairs in the training set** and **6 of the original 8,570 pairs in the testing set** for Deepfashion, while no data is filtered out for the MVP dataset. Such a ratio of test pairs (6/8,570) has a negligible effect on the overall performance. To validate this quantitatively, we report the result of our method and the closest competitor [4] (Pose with Style) on the full testing set of Deepfashion. The experimental results are reported in Table 2 in the supplemental material and as follows:
>
> | Method | mIoU (the higher the better) | LPIPS (the lower the better) | FID (the lower the better) |
> | :-: | :-: | :-: | :-: |
> | Pose with Style (filtered) | 62.74 | 0.1997 | 16.35
> | Pose with Style (full) | 62.49 | 0.1998 | 16.35
> | 3D-GCL (Ours, filtered) | 75.76 | 0.1725 | 10.58 |
> | 3D-GCL (Ours, full) | 75.91 | 0.1725 | 10.58 |
>
> It is apparent from the above results that the effects of the 6 filtered out test pairs can be ignored, and the proposed method is superior to the baseline method in both cases.
>
> Secondly, the closest competing baseline method [4] and also [3] heavily rely on the DensePose results and will thus benefit from this process as well, leading to a fair comparison.
>
>
> **Detail of HardPose**
> We first employ the calculating mechanism for pose complexity in [5] to filter out easy samples, then manually pick out the pairs that contain visually "Hard" posture from the testing set. Pairs with large viewpoint and hand position contrast are added into the testing set. Here we define "HardPose" as opposite to standard posture, which is described as face forward and hands down. We compute the complexity score (lower is more complex) of our final filtered subset, getting 74.39 for Deepfashion and 63.45 for MPV on the Hardset, compared to 77.87 for Deepfashion and 70.09 for MPV on the Fullset. This quantitative result is consistent with the picking rules and supports that our HardPose testing set is reliable. We also include some visual examples of our HardPose testing set as well as an easy to hard example in the supplementary.
>
>
> **Structure detail of Ablation study**
> To evaluate the performance of 3D-GCL w/o global correspondence (3D-GCL $\ast$), we replace the correlation module of Stage I in Figure 2 with the block that is demonstrated in Figure 3(a). In addition, the loss for the global correspondence $L_o$ is replaced with $L_f$ at resolution of 64 correspondingly.
>
> ---
>
> [1] ZFlow: Gated Appearance Flow-based Virtual Try-on with 3D Priors. ICCV 2021
>
> [2] Style-Based Global Appearance Flow for Virtual Try-On. CVPR 2022
>
> [3] Parser-Free Virtual Try-on via Distilling Appearance Flows. CVPR 2021
>
> [4] Pose with Style: Detail-Preserving Pose-Guided Image Synthesis with Conditional StyleGAN. ACM Transactions on Graphics 2021
>
> [5] Towards Photo-Realistic Virtual Try-On by Adaptively Generating↔Preserving Image Content. CVPR 2020

---

> > ### Comment · Reviewer_7wqS · 2022-08-08
> > **Thanks**
> >
> > Thanks for the authors' rebuttal. They addressed some of my concerns. I decide to increase my rating to borderline accept.

---

> ### Author Response · Authors · 2022-08-02
> **Response to Reviewer 7wqS Part 1**
>
> Thank you for your constructive suggestions and valuable feedback. We are pleased to hear that you appreciate both the qualitative and quantitative results produced by our method. In the following, we will address your concerns one by one.
>
> * **Missing comparisons**: Missing comparisons. Some recent works also focus on incorporating 3D priors [1] or global correspondence [2] into image-based virtual try-on. There is no deep discussion on the difference between this work and [1] [2]. Quantitative comparisons with [1][2] are also missing. Comparisons with the above works are needed to elucidate the main contributions of this work.
> * **Unfair experimental setup**: Unfair experimental setup. Line 224. "pairs without properly detected IUV maps or SMPL models are filtered out." I am concerned whether this is a fair comparison with other methods, as filtering out failure detections is beneficial for testing the proposed method. This makes the empirical results less convincing.
> * **Detail of HardPose**: What are the selection criteria for the HardPose test set? Is there a quantitative analysis of how hard the HardPose test set is compared to the public benchmarks?
> * **Structure detail of Ablation study**: Abalation study Table 3. What is the framework like for 3D-GCL w/o global correspondences?
>
> **Missing comparisons**
> Thank you for the suggestion. We will add the following discussion on the differences between our 3D-GCL and the mentioned approaches [1] and [2] in the paper:
>
> >While [1] also incorporates 3D priors during training, we argue that there exist intrinsic differences between [1] and our 3D-GCL, in terms of the intention and the derivation of the 3D prior. In [1], the 3D prior is introduced in the Segmentation-Assisted Dense Fusion (i.e., the try-on synthesis module) by taking the DensePose as input and reconstructing it in the output. This will facilitate the synthesis network to preserve
> structural and geometric integrity of the try-on results as mentioned in the original paper, but also means that the 3D prior in [1] is directly derived from the input DensePose. However, in our 3D-GCL, we innovatively employ the 3D prior to provide precise guidance to when learning the correspondence, which allows the warping module to preserve the garment texture even for challenging poses. Besides, the 3D prior of our 3D-GCL is derived from the 3D vertex correspondence between the SMPL model of the same person under various poses.
> >
> >On the other hand, although [2] proposes a global flow estimation module for garment deformation, it does not explicitly model the global correspondence between the source garment feature and the target pose feature. Specifically, [2] utilizes the style vector to modulate the weights of the StyleGAN-based network, where the style vector is obtained by concatenating the 1-D garment vector and the 1-D person vector.
> However, such a 1-D global style vector just provides the flow estimation network with the global information of the garment and person, rather than the global correspondence between the source and target feature.
> Instead, our 3D-GCL explicitly models the global correspondence between the garment and the person features by calculating the correspondence matrix in the low-resolution block and uses it as initial state for the high-resolution flow estimating blocks.
>
> We further have conduct a quantitative comparison with [2] during the rebuttal. For this we load their pretrained model and conduct experiments under the person-to-person setting on the testing set of Deepfashion. Note, we were unfortunately unable to re-implement and retrain the model ourselves due to the limited rebuttal time.
>
> | Method | mIoU (the higher the better) | LPIPS (the lower the better) | FID (the lower the better) |
> | :-: | :-: | :-: | :-: |
> | FS-VTON | 44.15 | 0.2420 | 31.64
> | Ours | **75.76** | **0.1725** | **10.58**
>
> [1] and [2] are both representatives for the garment-to-person line of research. In this work, however, we have instead chosen PF-AFN (PBAFN)[3] to represent this of research, which has been demonstrated to be a stronger baseline than ZFlow [1] (Table 1 in [2]), and otherwise focused more on the more relevant person-to-person comparisons.

---

### Official Review · Reviewer_AD5s · 2022-07-16

**Rating:** 7
**Confidence:** 5
**Soundness:** 3 good
**Presentation:** 3 good
**Contribution:** 3 good

**Summary:**

The paper present a 3D-aware Global Correspondence Learning to tackle the virtual try-on problem. The method injects 3D priors onto the feature learning step via SMPL-based approach. The method shows benefits on using such an approach on two widely used dataset by the community

**Questions:**

- If someone has a one-to-one correspondance from the DensePose, why would there be a need for the Stage I network . This is not to trivialize Stage I model, but rather to make sure that the simple waping filed obtained from DensePose are not enough and hence justifies its role. A baseline with only the one-to-one correspondance from the DensePose is much appreciated I think
- How the method deals with wrong semantic segmentation obtained from the human parser?

**Limitations:**

- As mentioned above, SMPL (DensePose) models are known to fail on hard posture especially on dance ones. How would the method deal on such cases? Relying solely on SMPL might give very wrong prediction, it would be good if the authors would develop on this

**Strengths And Weaknesses:**

**Strengths**
- The paper is well written and the ideas are easy to follow
- Conducted experiments on two datasets, DeepFashion and MPV
- Ablation study on the proposed 3D-GCL framework
- Good qualitative results on the try-on clothes

**Weaknesses**
- The model depends heavily on the SMPL model estimation.

---

> ### Author Response · Authors · 2022-08-02
> **Response to Reviewer AD5s**
>
> Thank you for your careful and comprehensive comments. We are glad to hear that you appreciate the idea of our 3D-GCL and the experiments conducted in the paper. In the following, we will address your concerns point by point:
>
> * **Dependency on SMPL model**: "The model depends heavily on the SMPL model estimation."
> * **Necessity of stage I**: "If someone has a one-to-one correspondance from the DensePose, why would there be a need for the Stage I network. This is not to trivialize Stage I model, but rather to make sure that the simple waping filed obtained from DensePose are not enough and hence justifies its role. A baseline with only the one-to-one correspondance from the DensePose is much appreciated I think."
> * **Influence of wrong segmentation:** How the method deals with wrong semantic segmentation obtained from the human parser?
> * **Influence of wrong pose prediction:** As mentioned above, SMPL (DensePose) models are known to fail on hard posture especially on dance ones. How would the method deal on such cases? Relying solely on SMPL might give very wrong prediction, it would be good if the authors would develop on this.
>
> **Dependency  on SMPL model**
> We believe that our model depends less heavily on the SMPL model compared to existing SMPL-based methods since we introduce the information through a learning-based approach. This is in contrast to recent methods [1,2], which directly add precomputed flows into the network pipeline, which causes heavy dependence on the SMPL model estimation. Our method instead learns the flow distribution which makes it possible for the network to correct the error introduced by outliers of the estimated human mesh distribution.
>
> **Necessity of stage I**
> It is correct that a simple warping flow obtained from DensePose is insufficient due to the limited scalability of the predefined UV space as we discuss in L106-L109.
> While we did not add such a baseline explicitly, one of the original baseline approaches [1] inpaints the warping flow obtained from DensePose and thus yields a similar comparison. Note, [1] also conducts additional ablation studies which show that directly using the Densepose flow will produce inferior results.
>
>
> **Influence of wrong segmentation**
> Most of the parser-based methods will fail without guidance of correct parsing results and our method is not an exception. We alleviate the influence of parsing errors by selecting the most popular human parsing prediction network [3].
> In this work, we focus on Virtual Try-on strategies for scenarios of diverse pose and viewpoint variations and empirically ignore the potential negative impact of wrong segmentation results.
> While such errors are ignored in consideration of our main focus, there do exist some parser-free solutions [4,5,6] particularly designed to handle parsing errors. One possible solution is to train a parser-free student model with our original pipeline by incorporating knowledge distillation strategies. We have elaborated this in our limitation section and further discussed the possible influence of parsing errors on our model.
>
> **Influence of wrong pose prediction**
> Problems caused by wrong pose estimation share similarity with parsing errors. Our method is unable to handle wrong pose input as it is not a trivial task to infer the target posture without guidance provided by a correct pose representation, e.g. keypoints, mesh, UV map. Since our main goal is to achieve Virtual Try-on, we assume that input pose representations sent into the network are reliable. Based on this assumption, we believe that our 3D-GCL is able to compute the global correspondence of the given poses extracted from source and target person and therefore facilitates garment warping under diverse scenarios.
> To tackle the difficulty of hard pose estimation, possible solutions are to fuse temporal information from a video or use knowledge distillation methods to exclude non-image inputs. We have added this discussion in the revised version of our paper.
>
> ---
>
> [1] Pose with Style: Detail-Preserving Pose-Guided Image Synthesis with Conditional StyleGAN. ACM Transactions on Graphics 2021
>
> [2] Dressing in the Wild by Watching Dance Videos. CVPR 2022
>
> [3] Graphonomy: Universal Human Parsing via Graph Transfer Learning. CVPR 2019
>
> [4] Do Not Mask What You Do Not Need to Mask: a Parser-Free Virtual Try-On. ECCV 2020
>
> [5] Parser-Free Virtual Try-on via Distilling Appearance Flows. CVPR 2021
>
> [6] Style-Based Global Appearance Flow for Virtual Try-On. CVPR 2022

---

### Author Response · Authors · 2022-08-02
**General Response -- thanks to all reviewers for constructive and insightful feedback**

We would like to thank all reviewers for their positive affirmations on the novelty and potential impact of this paper (e.g., Reviewer AD5s \& nqeh). With the proposed 3D global correspondence learning framework, our method outperforms existing virtual try on methods on two public datasets, especially on cases with hard postures (**agreed by all reviewers**). Besides, the writing of this paper is clear and easy to follow (Reviewer AD5s \& nqeh).

Following the constructive suggestions and comments of the reviewers, we have revised our paper and provided more experimental results to demonstrate the advantages of the proposed method against existing virtual try-on methods. Particularly, we have

1. explained our experimental setups (training/test splits) in detail (Reviewer bUCs) in Sec. 4.1 of the supplementary material;

2. included our quantitative criterion for identifying HardPose samples (Reviewer 7wqS \& bUCs) in Sec. 2 of the supplementary material;

3. verified that the number and effects of our filter-out pairs (i.e., without detected SMPL models/IUV maps) can be ignored (Reviewer 7wqS \& bUCs) in Sec. 6.1 of the supplementary material;

4. included extra state-of-the-art methods for comparison (Reviewer 7wqS) and more ablation study results (Reviewer nqeh) in the revised paper and supplementary material;


The main revisions in our paper and the supplemental material are marked in **RED**. We hope that our efforts address the concerns of all reviewers sufficiently.

---

### Meta-Review · Area_Chair_4LWN · 2022-08-28

**Recommendation:** Accept
**Confidence:** Certain

**Metareview:**

This paper received 4 positive reviews: 2xBA + WA+ A. All reviewers acknowledged that the proposed approach is simple and effective, it is well presented, and the claims are supported by strong empirical performance and extensive evaluation on several datasets. The remaining questions and concerns were addressed in the authors' responses, which seemed convincing to the reviewers. The final recommendation is therefore to accept.

**Award:**

No

---

### Decision · Program_Chairs · 2022-09-14

Accept